# GameInstruct: Teaching Machines to Reason via Chameleon Game

## Abstract

Self-play has emerged as a promising approach for generating alignment data to reduce the annotation costs during the alignment process. By introducing specific game rules and utilizes the model's own language capabilities to generate data samples, self-play has achieved promising results. However, traditional self-play methods face two major challenges: insufficient data diversity during self-iterative training and difficulties in reward signal design. To solve these problems, this paper introduces GameInstruct, a complex multi-player adversarial environment that increases the complexity of self-play generated data during self-iterative training. Specifically, we employ the "Chameleon Game", where interactions between multiple players raise the diversity of the generated data, improving the model's reasoning abilities, Additionally, we further propose a dynamic reward algorithm to capture signals within player conversations during the whole game. Experimental results show that compared to existing self-play methods, GameInstruct achieves significant improvements on the HuggingFace Open-LLM-Leaderboard reasoning benchmark while demonstrating continuous improvement and increasing data diversity during self-iterative training.

## 1 Introduction

Alignment aims to ensure the model responds according to the user's goals and needs, which is primarily achieved through learning from human-annotated instruction data and preference data(Ji et al., 2024). However, the high cost of human annotated data hinders the way to develop stronger LLMs efficiently. Additionally, with the continuous development of strong LLMs such as GPT-4 (OpenAI, 2023) and Gemini (Gemini, 2024), human face challenges when supervising responses of LLMs to keep in consistency with their preferences (Burns et al., 2023).

To solve above problems, self-play (Cheng et al., 2024; Chen et al., 2024) emerges as a particularly promising approach to generate alignment data. Self-play usually utilizes the intrinsic language capability of LLMs by introducing specific game rules and generate relevant samples to improve LLMs. Numerous self-play methods have been proposed these days, among which generator-discriminator type (Cao et al., 2024) has been widely studied. SPIN (Chen et al., 2024) introduces game from the idea of GAN (Goodfellow et al., 2014), where discriminators distinguish LLMs' responses from the golden while generators generate responses as close to the golden as possible. Another line of works is attacker-defender type, where attacker and defender share inverse objective. SPAG (Cheng et al., 2024) utilizes Adversarial Taboo (Yao et al., 2020), where attackers trick the defenders to answer specific taboo word while defenders guess the taboo word. Both of these works have achieved satisfactory outcomes.

However, traditional self-play methods face two major challenges: *poor diversity during self-iterative training* and *the difficulty of reward signal design*. The first challenge arises when models overly rely on self-generated data, which often lacks diversity and complexity. This lack of diversity and complexity limits the model's ability to explore broader data patterns. As noted by Shumailov et al. (2024b), this reliance can lead LLMs to drift from the original data distribution, eventually resulting in model collapse. The second challenge lies in the design of effective reward signals. Traditional methods typically focus on game outcomes rather than leveraging the rich information embedded in conversations to guide the training process. This is especially important in large models, where focusing solely on final outcomes can cause LLMs to overlook critical intermediate

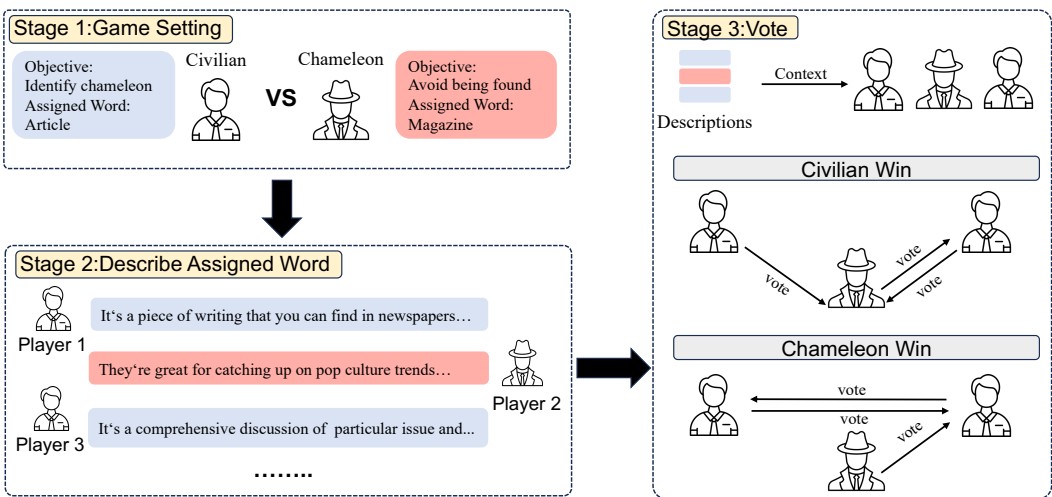

Figure 1: An example of Chameleon Game. This game involves chameleons and civilians engaging in a iterative interaction, where they are assigned specific words and required to describe words from multiple aspects. Subsequently, they are asked to make votes, where chameleons win if they are not voted, otherwise civilians win.

reasoning steps. As a result, LLMs may adopt suboptimal or inefficient strategies (Zhai et al., 2022). Our experiments in Section 4 demonstrate that using reward signals based solely on game results significantly hinders the reasoning capabilities of LLMs.

To address this issue, this paper introduces GAMEINSTURCT, a complex multi-player adversarial environment that increases the complexity of self-play generated data during training. The core idea of GAMEINSTURCT is to use multi-player interactions and a diversified data generation process to prevent the model from getting stuck in repetitive self-play, reducing the risk of model collapse (Shumailov et al., 2024b). Specifically, we utilize the "Chameleon Game" (Xu et al., 2023), as described in Fig 1, where civilians are given the same word and attempt to identify the chameleon, while the chameleon avoids being identified. This game requires strong skills of expression, deception, and reasoning, and the ease of collecting numerous assigned words enables a wide range of topics, ensuring the diversity of the generated data. Besides, GAMEINSTRUCT introduces a dynamic reward learning algorithm to capture signals within player conversations, incorporating interactions between players into reinforcement learning. This allows the model to receive continuous and targeted feedback throughout the entire game, optimizing performance progressively and avoiding the limitations of relying solely on final outcomes.

To verify the effectiveness of GAMEINSTURCT, we conduct experiments on multiple open-source LLMs across different training iterations and sampling temperatures. The experimental results demonstrate a significant improvement over state-of-the-art self-play methods on the reasoning benchmark, HuggingFace Open-LLM-Leaderboard (Beeching et al., 2023). Furthermore, our experiments indicate that GAMEINSTRUCT reduces the risk of model collapse and maintains continuous improvement, with increased data diversity during the self-iterative training.

Generally, our contribution can be summarized as follows:

- We introduce a training method called GAMEINSTURCT. By introducing a complex multi-player environment, GAMEINSTURCT enhances LLM interactions and generates more diverse data, resulting in significant improvement in their reasoning capabilities.

- We propose a reward assigning method named *dynamic reward*. By assigning reward based on the descriptions of each players, *dynamic reward* better captures signals within player conversations, further improving the reasoning abilities of LLMs.

- We verify the effectiveness and robustness of GAMEINSTURCT, showing improvement in reasoning and a continuous improvement along during self-iterative learning in comparison with the state-of-the-art self-play methods.

Figure 2: The overall training process of GAMEINSTURCT: (1) Imitation learning from game data generated by GPT. (2) Assigning reward of LLMs generated game data. (3) Reinforcement learning the new LLMs on reward data.

## 2 PRELIMINARY

Reinforcement learning (RL) has played an increasingly important role in language model training (Ouyang et al., 2022a; Ramamurthy et al., 2023). Among the existing RL methods, reinforcement learning from human feedback (RLHF) has gained more and more attention (Yuan et al., 2023; Ouyang et al., 2022b). RLHF first learns a reward model from the human annotated preference pairs, and then optimizes the policy model in order to maximize the expected reward:

$$\mathcal{L}_{\text{RLHF}}(\pi_\theta) = -\mathbb{E}_{\boldsymbol{x} \sim D, \boldsymbol{y} \sim \pi_\theta(\boldsymbol{y}|\boldsymbol{x})}[r(\boldsymbol{x}, \boldsymbol{y})] \tag{1}$$

In order to optimize the above reward, proximal policy optimization (PPO) algorithm (Schulman et al., 2017b) has been introduced as a solution:

$$\mathcal{L}_{\text{PPO}}(\pi_\theta) = -\mathbb{E}_{\boldsymbol{x} \sim \mathcal{D}, \boldsymbol{y} \sim \pi_{\bar{\theta}}(\boldsymbol{y}|\boldsymbol{x})} \left[ \frac{\pi_\theta(\boldsymbol{y}|\boldsymbol{x})}{\pi_{\bar{\theta}}(\boldsymbol{y}|\boldsymbol{x})} \hat{A}^{\pi_{\bar{\theta}}} - \beta \text{KL}[\pi_{\bar{\theta}} \| \pi_\theta] \right] \tag{2}$$

However, PPO for LLMs has been continually challenged due to its inefficient online sampling and the unstable training outcomes. Among the improvements versions of RL, direct policy optimization (DPO) (Rafailov et al., 2024) has been proposed as a strong RL algorithm, where preference pairs are needed to optimize the policy model instead of the reward of specific preference data.

$$\mathcal{L}_{\text{DPO}}(\pi_\theta; \pi_{\text{ref}}) = -\mathbb{E}_{(x, y_w, y_l) \sim \mathcal{D}} \left[ \log \sigma \left( \beta \log \frac{\pi_\theta(y_w \mid x)}{\pi_{\text{ref}}(y_w \mid x)} - \beta \log \frac{\pi_\theta(y_l \mid x)}{\pi_{\text{ref}}(y_l \mid x)} \right) \right]. \tag{3}$$

## 3 METHODOLOGY

This section mainly introduces the overall training process of GAMEINSTURCT. Specifically, Section 3.1, Section 3.2 and Section 3.3 introduce the game process, the theoretical analysis of chameleon game and the imitation learning from GPT-4, indicating the significance and motivation of such multi-player game. Subsequently, Section 3.4 and Section 3.5 illustrate the assigning process of dynamic reward and its application in reinforcement learning, representing the essence of such method. The overall process of GAMEINSTRUCT is presented in Fig 2

### 3.1 CHAMELEON GAME INTRODUCTION

The Chameleon game applied in LLMs is first introduced by Xu et al. (2023), in which civilians and chameleons involve an multi-turn interaction. The game consists of multiple players, divided into civilians and chameleons, where civilians share a target word, and the chameleons are assigned a different word. Each round players are asked to describe the assigned word. After two rounds of description, all players are required to vote. If the chameleons receive more votes, the civilians win, otherwise the chameleon wins. The example of Chameleon game is in Figure 1.

The goal of chameleons is to avoid being voted out, where they should pay attention to civilians' description related to the word and deduce their assigned word or any topics related to it and describe. In contrast, civilians are required to identify their own teammate and vote out the chameleon. Thus, civilians need to introduce their assigned words as exact as possible to find other civilian but bring less information to the actual chameleon as possible.

## 3.2 CHAMELEON GAME MODELING

We identify the Chameleon Game as a multi-player zero-sum Markov game, which can be described by $(\mathcal{S}, \mathcal{A}, \mathcal{F}, \mathcal{R})$ [1]:

- State space: $\mathcal{S} = \{s_t, s'_t, s''_t \mid 1 \leq t \leq T\}$ indicates the history conversation of Chameleon Game, which contains three types of states. $s_t = (w, u_1, v_1, u'_1, u_2, \ldots, u_t)$, $s'_t = (w, u_1, v_1, u'_1, u_2, \ldots, u_t, v_t)$ and $s''_t = (w, u_1, v_1, u'_1, u_2, \ldots, u_t, v_t, u'_t)$ where $u_t, u'_t$ and $v_t$ are the utterances of civilians and chameleon, respectively[2]. States $s_t, s'_t$ and $s''_t$ end with utterances $u_t, v_t$ and $u'_t$ from civilians and chameleons, respectively.

- Action: $\mathcal{A}$ marks the possible action space for LLMs to choose, which is equivalent to the possible token sequence space of specific LLMs, indicating the generation process of LLMs.

- Transition Function $\mathcal{F} : \mathcal{S} * \mathcal{A} = \mathcal{S}$ deterministically appends the utterance $u_i, u'_i$ or $v_i$ at the end of the dialogue, and converts $s'_t = F(s_t, v_t)$, forming a new game history.

- Reward $r : \mathcal{S} \times \mathcal{A} \rightarrow \mathbb{R}$ evaluates actions based on their corresponding states with rewards $r(s, v)$, $r(s', u')$ and $r(s'', u)$, respectively. Specifically, $\mathbb{R}$ marks the reward of each generation process of LLMs based on the game history.

In this game, we denote $u, u'$ and $v$ as the civilian's policy and the chameleon's policy. Each game episode can be regarded with the following probability:

$$P(t) = P(s_0) \prod_{t=1}^{T} P(s_t | s''_t) \prod_{t=1}^{T} P(s'_t | s_{t-1}) \prod_{t=1}^{T} P(s''_t | s'_t) =: (\mu \times \nu \times \mu'), \tag{4}$$

where $P(s_0)$ marks the initial state of the game, indicating the game rules and assigned word. With the theory above, the overall objective of the Chameleon Game is:

$$\max_{\nu} \min_{\mu, \mu'} \mathcal{L}_{\text{Game}}(\mu, \nu, \mu') := \mathbb{E}_{t \sim \mu \times \nu \times \mu'} [R(t)], \tag{5}$$

where chameleons try to maximize their reward $R(t)$, which is identical to the total reward by optimizing policy $\nu$, and civilians search for policies $\mu$ and $\mu'$ to maximize their reward $-R(t)$, which happens to minimize total reward $R(t)$.

To play the game successfully with LLMs, we design game rule prompts as well as diverse actor prompt templates $f_{\text{cha}}$ and $f_{\text{civ}}$ for the chameleons and civilians respectively. Thus Game policies for both players are as followed:

$$\mu'_\theta(u'|s') = \pi_\theta(u'|f_{\text{civ}}(s')), \mu_\theta(u|s'') = \pi_\theta(u|f_{\text{civ}}(s'')), \nu_\theta(v|s) = \pi_\theta(v|f_{\text{cha}}(s)), \tag{6}$$

where $\pi_\theta$ stands for the specific LLM and $\mu_\theta, \mu'_\theta, \nu_\theta$ indicate the game policy for LLMs with different roles to learn.

## 3.3 IMITATION LEARNING

Since current open-source LLMs, especially small size ones are unable to fully follow complex game rules strictly in $f_{\text{cha}}(s)$ and $f_{\text{civ}}(s')$, generating low quality data. We introduce imitation learning using GPT-4 (OpenAI, 2024) to ensure that the LLMs outputs align with the game rules. We prompt GPT-4 with different players as civilians and chameleons and ask them to play the Chameleon Game to collect the data from them, as Figure 1 described.

After gathering the imitation data $\mathcal{D}_{\text{im}}$ from GPT-4, we divide it into an civilians-winning set $\mathcal{D}_{\text{im}}^{\text{civ}}$ and a chameleons-winning set $\mathcal{T}_{\text{im}}^{\text{cha}}$. In order to help LLMs understand the rule of chameleon game, we design the loss similar to instruction tuning process, where the log-likelihood of the model's responses during the conversation is maximized. We adopt the imitation learning idea from Cheng et al. (2024) and set a penalty preventing LLMs from overfitting to the game, as the following:

$$\mathcal{L}_{\text{im}}^{\text{cha}}(\pi_\theta) = -\mathbb{E}_{t \in \mathcal{T}_{\text{im}}^{\text{cha}}} \left[ \frac{1}{T} \sum_{t=1}^{T} \log \pi_\theta(v_t | f_{\text{cha}}(s_t)) + \beta_1 \text{KL}[\pi_\theta \| \pi_{\text{ref}}] \right], \tag{7}$$

---

[1] Applying three-player scenario for simplicity

[2] Chameleon is set to describe the word in the second order in this case

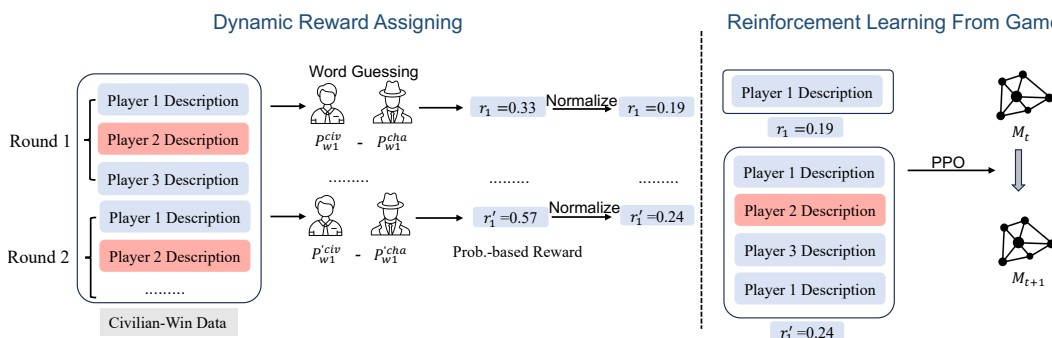

Figure 3: Process of dynamic reward and reinforcement learning: 1)Players are prompted to calculate the output probability of winners' assigned word, which is applied for reward assigning. 2) Game Data with reward is utilized for PPO training to acquire the new LLM.

$$\mathcal{L}_{\text{im}}^{\text{civ}}(\pi_\theta) = -\frac{1}{2}\mathbb{E}_{\boldsymbol{t}\in\mathcal{T}_{\text{im}}^{\text{civ}}}\left[\frac{1}{T}\sum_{t=1}^{T}\log\pi_\theta(\boldsymbol{u}_t|f_{\text{civ}}(\boldsymbol{s}_t'')) + \beta_1\text{KL}[\pi_\theta\|\pi_{\text{ref}}]\right]$$
$$-\frac{1}{2}\mathbb{E}_{\boldsymbol{t}\in\mathcal{T}_{\text{im}}^{\text{civ}}}\left[\frac{1}{T}\sum_{t=1}^{T}\log\pi_\theta(\boldsymbol{u}_t'|f_{\text{civ}}(\boldsymbol{s}_t')) + \beta_1\text{KL}[\pi_\theta\|\pi_{\text{ref}}]\right], \tag{8}$$

where the coefficient $\beta_1$ and the Kullback-Leibler divergence $\text{KL}[\pi_\theta\|\pi_{\text{ref}}]$ prevents the LLMs from over-fitting on the game and maintains the LLMs' general abilities. The reference model $\pi_{\text{ref}}$ marks the LLMs before imitation learning process. Thus, the overall imitation learning objective is as followed.

$$\mathcal{L}_{\text{im}}(\pi_\theta) = \frac{1}{2}\mathcal{L}_{\text{im}}^{\text{civ}}(\pi_\theta) + \frac{1}{2}\mathcal{L}_{\text{im}}^{\text{cha}}(\pi_\theta) \tag{9}$$

## 3.4 DYNAMIC REWARD ASSIGNING

To better capture the information between conversations of players, we propose to use dynamic reward during reinforcement learning by applying the policy model as judge to assign reward to the conversation between players. By assigning each description with a certain reward, information contained in the conversation is better captured compared with splitting the global reward to each sentence with regard to the conversation turns (Cheng et al., 2024).

The commonly employed method of global reward allocation, which relies on the outcome of each episodic, fails to effectively capture reward signals across dialogues (Cheng et al., 2024), since such an approach is grounded in a strong assumption that rewards between each dialogue follow a zero-sum pattern. However, the zero-sum nature of the overall game does not necessarily imply such a conclusion of each conversation turns. Adopting this method directly may result in the loss of important reward signals during intermediate stages.

For each state, the chameleon reward and civilian reward in the total game should be summed to zero, so that the game is zero-sum:

$$\sum_{t=1}^{T} r(\boldsymbol{s}_{t-1}'', \boldsymbol{u}_t) + \sum_{t=1}^{T} r(\boldsymbol{s}_t, \boldsymbol{v}_t) + \sum_{t=1}^{T} r(\boldsymbol{s}_t', \boldsymbol{u}_t') = 0. \tag{10}$$

In order to better capture the information in the conversation, we send response of each players to subsequent players for word guessing. To be more specific, we prompt the guessing player with word-guess prompt and acquire the probability of the assigned word, allowing them to guess the probability that this description is the assigned word, just as illustrated in Figure 3

It's worth mentioning that chameleons and civilians share a reverse objective, as a result, the probability of assigned word plays a distinct role in reward of each action. For civilians, the primary goal

is to identify their own teammates and avoid being detected by the chameleon. Thus we set

$$r(\boldsymbol{s}'_t, \boldsymbol{u}'_t) = P_{civ}(\boldsymbol{u}'_t, w_{civ}) - P_{cha}(\boldsymbol{u}'_t, w_{civ}) \tag{11}$$

where $P_{civ}$ and $P_{cha}$ represent the probability of civilians and chameleon guess the assigned words from $\boldsymbol{u}'_t$ respectively, while $w_{civ}$ represents the assigned word of civilians.

As for chameleons, the primary goal is to hide their true identity while tricking the civilians to identify themselves as teammates.

$$r(\boldsymbol{s}_t, \boldsymbol{v}_t) = P_{civ}(\boldsymbol{v}_t, w_{civ}) - P_{civ}(\boldsymbol{v}_t, w_{cha}) \tag{12}$$

We heuristically design reward to ensure that $R = 1$ if chameleon wins, $R = -1$ if the civilians win. After acquiring the total game result, we will perform softmax normalization on the rewards of different rounds of conversations with the same character to reassign them, thus obtaining the reward result for each state in the end.

### 3.5 Reinforcement Learning From Chameleon Game

With a group of episodes assigned with reward, we consider updating the chameleon policy $v_\theta$ with respect to the overall objective. The corresponding policy gradient for the chameleon is as followed

$$\nabla_\theta \mathcal{L}_{\mathrm{AG}}(\mu_\theta, \nu_{\bar{\theta}}, \mu'_\theta) = \mathbb{E}_{\boldsymbol{t} \sim \nu_{\bar{\theta}} \times \nu_{\bar{\theta}} \times \nu'_\theta} \Big[ \sum_{t=1}^{T} A_t^{\nu_{\bar{\theta}}} \cdot \frac{\nu_\theta(\boldsymbol{v}_t|\boldsymbol{s}_{t-1})}{\nu_{\bar{\theta}}(\boldsymbol{v}_t|\boldsymbol{s}_{t-1})} \cdot \nabla_\theta \log \nu_\theta(\boldsymbol{v}_t|\boldsymbol{s}_{t-1}) \Big], \tag{13}$$

where $A_t^{\nu_{\bar{\theta}}} = A^{\nu_\theta}(\boldsymbol{s}_{t-1}, \boldsymbol{v}_t)$ is the advantage of action $\boldsymbol{v}_t$. To estimate the expectation of $\nu_\theta(\boldsymbol{v}_t|\boldsymbol{s}_{t-1})$ unbiased, We apply importance sampling following TRPO (Schulman et al., 2017a) with its original sampling strategy $\nu_{\bar{\theta}}(\boldsymbol{v}_t|\boldsymbol{s}_{t-1})$, which assists in approximating the expected reward of $\nu_\theta$.

Inspired by PPO (Schulman et al., 2017b), we apply the following loss to optimize $\mathcal{L}_{\mathrm{AG}}(\mu_\theta, \nu_{\bar{\theta}}, \mu'_\theta)$:

$$\mathcal{L}_{\mathrm{game}}^{\mathrm{cha}}(\pi_\theta) = -\mathbb{E}_{\boldsymbol{t} \in \mathcal{T}_{\bar{\theta}}} \Big[ \sum_{t=1}^{T} \frac{\nu_\theta(\boldsymbol{v}_t|\boldsymbol{s}_{t-1})}{\nu_{\bar{\theta}}(\boldsymbol{v}_t|\boldsymbol{s}_{t-1})} \hat{A}_t^{\nu_{\bar{\theta}}} - \beta_2 \mathrm{KL}[\pi_\theta \| \pi_{\bar{\theta}}] \Big], \tag{14}$$

Where the regularize $\mathrm{KL}[\pi_\theta \| \pi_{\bar{\theta}}]$ and the coefficient $\beta_2$ controls the training process of reinforcement learning, keeping the training model in specific distance with it in last iteration. Following PPO (Schulman et al., 2017b), we apply empirical estimation of advantage $\hat{A}_t^{\nu_{\bar{\theta}}}$ of former policy to approximate the advantage $A_t^{\nu_\theta}$. In this case, we apply the reward of each sentence as the advantage. Similarly, from the perspective of the chameleons , the corresponding loss is :

$$\mathcal{L}_{\mathrm{game}}^{\mathrm{cha}}(\pi_\theta) = -\mathbb{E}_{\boldsymbol{t} \in \mathcal{T}_{\bar{\theta}}} \Big[ \sum_{t=1}^{T} \frac{\pi_\theta(\boldsymbol{v}_t|f_{\mathrm{cha}}(\boldsymbol{s}_{t-1}))}{\pi_{\bar{\theta}}(\boldsymbol{v}_t|f_{\mathrm{cha}}(\boldsymbol{s}_{t-1}))} \hat{A}_t^{\nu_\theta} - \beta_2 \mathrm{KL}[\pi_\theta \| \pi_{\bar{\theta}}] \Big]. \tag{15}$$

In fact, the above two loss functions can be combined together to illustrate the actual improvements that LLMs will gain during this game from the perspectives of both civilians and chameleon. Therefore, we can obtain the following overall loss in GAMEINSTURCT as followed:

$$\mathcal{L}_{\mathrm{game}}(\pi_\theta) = -\frac{1}{2}\mathcal{L}_{\mathrm{game}}^{\mathrm{cha}}(\pi_\theta) - \frac{1}{2}[\frac{1}{2}\mathcal{L}_{\mathrm{game}}^{\mathrm{civ_1}} + \frac{1}{2}\mathcal{L}_{\mathrm{game}}^{\mathrm{civ_2}}] - \alpha\mathbb{E}_{(\boldsymbol{x},\boldsymbol{y}) \sim \mathcal{D}_{\mathrm{IT}}}[\log \pi_\theta(\boldsymbol{y}|\boldsymbol{x})] \tag{16}$$

where $\mathcal{L}_{\mathrm{game}}^{\mathrm{civ_1}}$ and $\mathcal{L}_{\mathrm{game}}^{\mathrm{civ_2}}$ stand for the loss of two civilians respectively, which is similar to chameleon. $\mathbb{E}_{(\boldsymbol{x},\boldsymbol{y}) \sim \mathcal{D}_{\mathrm{IT}}}[\log \pi_\theta(\boldsymbol{y}|\boldsymbol{x})]$ stands also as a penalty during the reinforcement learning, which calculates the log-likelihood on specific Instruction Tuning dataset to prevent LLMs from overfitting the game.

## 4 Experiment

In this section, in order to verify the effectiveness and stability of GAMEINSTURCT in advancing reasoning ability of LLMs, we compared GAMEINSTURCT with traditional training methods and other self-play methods on reasoning benchmarks. The following are the details and results of our experiment:

Table 1: Results of baselines and GAMEINSTURCT Open-LLM-Leaderboard.

| Method | AVG | MMLU | ARC | Hellaswag | TruthfulQA | Winogrande | GSM8K |
|---|---|---|---|---|---|---|---|
| *Llama-3-8B as the base model* | | | | | | | |
| SFT | 66.87 | 66.95 | 60.32 | 78.41 | 51.22 | 74.38 | 68.50 |
| DPO | 67.12 | 67.41 | 60.95 | 78.96 | 51.76 | 74.70 | 68.81 |
| SPIN | 68.10 | 68.43 | 61.88 | 79.92 | 53.15 | 75.63 | 69.48 |
| SPAG | 67.50 | 67.28 | 62.35 | 78.84 | 52.03 | 75.79 | 68.73 |
| GAMEINSTURCT | **69.09** | **68.70** | **63.25** | **80.84** | **53.91** | **76.81** | **69.92** |
| *Llama-3-70B as the base model* | | | | | | | |
| SFT | 77.87 | 80.11 | 71.33 | 85.62 | 61.77 | 82.88 | 85.48 |
| DPO | 78.32 | 80.82 | 71.86 | 85.77 | 62.39 | 83.10 | 85.67 |
| SPIN | 79.88 | 82.35 | 72.61 | 86.85 | 63.64 | 84.73 | **89.07** |
| SPAG | 78.55 | 80.19 | 72.04 | 85.11 | 62.97 | 83.19 | 84.26 |
| GAMEINSTURCT | **80.47** | **82.53** | **73.97** | **87.77** | **64.35** | **85.19** | 89.03 |

## 4.1 EXPERIMENT SETTING

**Training Data**   The training data consists of the following parts:

- Assigned Words: We aim to play the Chameleon game with an extensive range of words so that diverse topics can be discussed during the self-play processes, which helps maintain the generalization ability of LLMs. Hence, we select word in relations from the Wordnet (Miller, 1992) as the target word-pairs.

- Imitation Learning Data: To enable the instruction-following ability of open-source LLMs on game rules, we use the same data collection process in Figure 1 via GPT-4 (OpenAI, 2024) API and play the Chameleon game one episode per target word. Due to the resource limitation, we only collect the GPT-4 self-play samples with the top 30K words.

- SFT&DPO Training Data: We apply UltraFeedback (Cui et al., 2024) as a baseline for comparison, to be more specific, we apply the query and chosen part of the preference dataset for SFT training stage, while the original preference dataset for DPO training.

**Evaluation**

- Reasoning Benchmark: To test the reasoning ability of LLMs, we consider applying Open-LLM-Leaderboard (Beeching et al., 2023), including HellaSwag (Zellers et al., 2019), ARC easy (ARC-e) & challenge (ARC-c) (Clark et al., 2018), WinoGrande (Sakaguchi et al., 2019), GSM8K (Cobbe et al., 2021), TruthfulQA (Lin et al., 2022) and MMLU (Hendrycks et al., 2021). TruthfulQA necessitates that the generated answers match exactly, while other benchmarks are primarily in a multiple-choice format. The details of these benchmarks are presented in Appendix.

- Diversity Metric: To test the diversity of the generated data from distinct self-play methodsWe apply Self-BLEU (Zhu et al., 2018) as the diversity metric, where higher Self-BLEU scores means poor diversity of data. To better represent the change of Self-BLEU over self-iterative training, we calculate the growth rate of Self-BLEU from the first training iteration. The details of this metric are presented in Appendix.

**Baselines**   We apply Supervised Fine-tuning(SFT) as well as the DPO for the initial comparison. Considering other self-play methods, we apply SPAG (Cheng et al., 2024), where LLMs play adversarial taboo game (Yao et al., 2020) by attacker tricking defender to output specific target words while they guess what the taboo words is. We also apply SPIN (Chen et al., 2024), where the objective of the training LLM is to identify the generation of their last-training version from the human generated sentence.

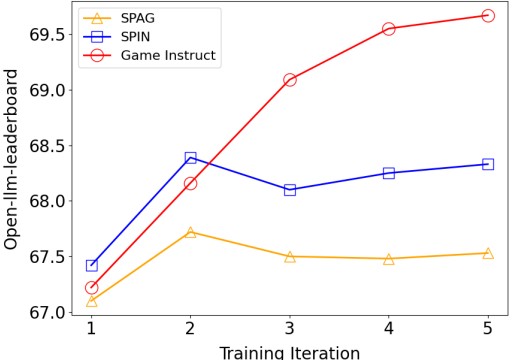

Figure 4: Open-LLM-Leaderboard results (AVG) on Llama3-8b training in distinct self-play methods throughout the entire iteration.

| Method | 0.5 | 0.7 | 0.9 |
|---|---|---|---|
| SPAG | 67.38 | 67.50 | 67.49 |
| SPIN | 68.01 | 68.10 | 68.35 |
| GAMEINSTURCT | **68.83** | **69.09** | **68.88** |
| w/o dynamic reward | 68.40 | 68.32 | 68.24 |

Table 2: Open-LLM-Leaderboard results (AVG) on Llama3-8b training in distinct self-play methods with diverse sampling temperature.

**Training Details**  For imitation learning, we set the learning rate as 5e-6, while for GAMEIN-STURCT training, the learning rate is 2e-6 and the hyperparameter $\alpha$ is set to 0.4. For the SFT baseline, we set the learning rate to 2e-6. As for the DPO baseline, we set the learning rate to be 3e-5. Among all training stages, the batch size is 128 and the max sequence length is 2048. Each training iteration of Self-play maintains one epoch over the offline collected trajectories. All our experiments are conducted using 64 NVIDIA A100 GPUs with 80GB memory.

## 4.2 MAIN RESULTS

To verify the overall results of GAMEINSTURCT, we conduct the experiments over six reasoning datasets compared with other baselines and their performance trend during the self-iterative training procedure, as presented in Table 1 and Figure 4, reveal that:

**1) GAMEINSTURCT exhibits a noteworthy improvement in advancing LLMs reasoning ability.** Our method demonstrates a stable improvement in multiple LLMs over the Open-LLM-Leaderboard as illustrated in Table 1 Specifically, for Llama3-8B, our method has improved 1.97% in average performance compared with basic alignment method such as DPO. While in comparison with state-of-the-art self-play method, GAMEINSTURCT still has its performance steadily improved by 0.99%, with 2.11% and 1.92% improvement in commonsense reasoning benchmark such as Winogrande and Hellaswag. As for strong LLM such as Llama3-70B, the effectiveness of GAMEINSTURCT preserves as well, with a rise of 0.59% in average performance on Open-LLM-Leaderboard compared with SPIN. We believe that such a improvement benefits both from multiplayer game increasing the complexity of the game and dynamic reward capturing more information embedded in the conversation data, which we will verify during ablation study.

**2) GAMEINSTURCT shows a trend of continuous improvement in reasoning throughout the training iterations.**  Figure 4 illustrates the average performance of distinct self-play methods of Llama3-8b on Open-LLM-Leaderboard. We find that our method keeps improving after two training iterations while other methods cease to improve and even degrade. Besides, even if GAME-INSTURCT has a relatively low performance during the first training iteration, GAMEINSTURCT outperforms SPIN and SPAG as the self-iterative training proceeds. At the fifth training iterations, both SPIN and SPAG gradually converge while GAMEINSTURCT keeps improving its performance. As Alemohammad et al. (2023) indicates that training with synthetic data, it is common that models performance converges or even degrades as the training proceeds, while our method sustains its performance over the training iterations. This is likely because other training methods begin to reach their equilibrium early in the training thus generate data in low diversity, while our method introduces complexity to the game and are more difficult to achieve equilibrium, which, ensures data diversity within the continuous cycle of training generation.

Table 3: Effect of dynamic reward in GAMEINSTURCT on Llama3-8B

| Iteration | dynamic reward | AVG | MMLU | ARC | Hellaswag | TruthfuQA | Winogrande | GSM8k |
|-----------|----------------|-----|------|-----|-----------|-----------|------------|-------|
| Iteration 1 | ✓ | 67.22 | 67.54 | 60.92 | 78.85 | 51.94 | 75.24 | 68.82 |
| | ✗ | 66.70 | 67.08 | 60.19 | 78.57 | 51.75 | 74.37 | 68.26 |
| Iteration 2 | ✓ | 68.19 | 68.47 | 62.08 | 80.02 | 52.78 | 76.26 | **69.98** |
| | ✗ | 67.55 | 67.83 | 61.35 | 79.27 | 52.43 | 75.79 | 69.07 |
| Iteration 3 | ✓ | **69.09** | **68.70** | **63.25** | **80.84** | **53.91** | **76.81** | 69.92 |
| | ✗ | 68.32 | 68.03 | 62.46 | 80.06 | 53.40 | 76.02 | 69.01 |

## 4.3 ABLATION STUDY

To further validate the effectiveness of multi-player game and dynamic reward, we calculate the diversity of generated data with distinct self-play methods during self-iterative training, as presented in Figure 5. And we also conduct experiment on GAMEINSTURCT without dynamic reward. Additionally, we calculate the performance of distinct self-play methods in several sampling temperature, as presented in Table 2 and Table 3, illustrate that:

**1) Dynamic reward steers the reasoning ability of LLMs as well as the stability of such improvement along with training.** From Table 2, we find that along with the training process of GAMEINSTURCT, LLMs reasoning ability consistently outperforms those without dynamic reward, with an average improvement of around 0.5%. Among these reasoning benchmark, LLMs' performance on GSM8k suffers the most from the absence of dynamic reward, indicating that dynamic reward is positive for LLMs to improve their capabilities on reasoning. While performance on TruthfulQA experience tiny performance distrubances. It is possible that TruthfulQA does not involve complex reasoning, but rather common sense questions and answers, and relies more on the LLMs knowledge rather than reasoning ability. Additionally, dynamic reward sustains its improvement and even becomes increasingly essential as the training iteration proceeds, with the performance gap in each iteration expanding continuously.

**2) GAMEINSTURCT exhibits strong robustness over sampling temperature.** The improvement of our method is steady across distinct setting of sampling temperature. Results in Table 2 indicate that GAMEINSTURCT maintains the improvement over other self-play methods in distinct temperature settings. Importantly, GAMEINSTURCT without dynamic reward tend to yield worse results, validating the efficacy of dynamic reward integration in enhancing the training outcomes for language models.

**3) GAMEINSTRUCT generates data with higher diversity and robustness against the training iterations.** We can conclude from Fiigure 5 that as the training proceeds, GAMEINSTRUCT generates the most diversified data compared with other self-play methods, with the lowest Self-BLEU scores across the whole training iterations. Besides, data generated by GAMEINSTRUCT is more robust across all these self-play methods, since the growth rate of Self-BLEU in GAMEINSTRUCT face the least improvement, representing a more stable trend during the self-iterative training.

## 5 RELATED WORK

**Self-play** This approach erefers to a paradigm where an agent learns by iteratively playing games against the replica of itself, thereby facilitating an escalating degree of challenge and complexity in the learning environment. It serves as the foundation of many successful specialzed AI system like AlphaGo Zero (Silver et al., 2017) , which demonstrated exceptional performance against human players. A Subsequent research has expanded upon the concept of self-play, exploring various adaptations and implementations (Chen et al., 2024, Cheng et al., 2024,Zheng et al., 2024,Wu et al., 2024). These self-play methods are mainly categorized in Generator-Discriminator type and Debate type (Cao et al., 2024). One of the former type is SPIN (Chen et al., 2024), which adapts the idea of GAN (Goodfellow et al., 2014) by introducing LLMs themselves to play generator, which distinguish generated data from the golden, and discrimnator, which generate data close to golden, at the same time. Expanding on this, Shaikh et al. (2024) incorporates replay comparison signals between

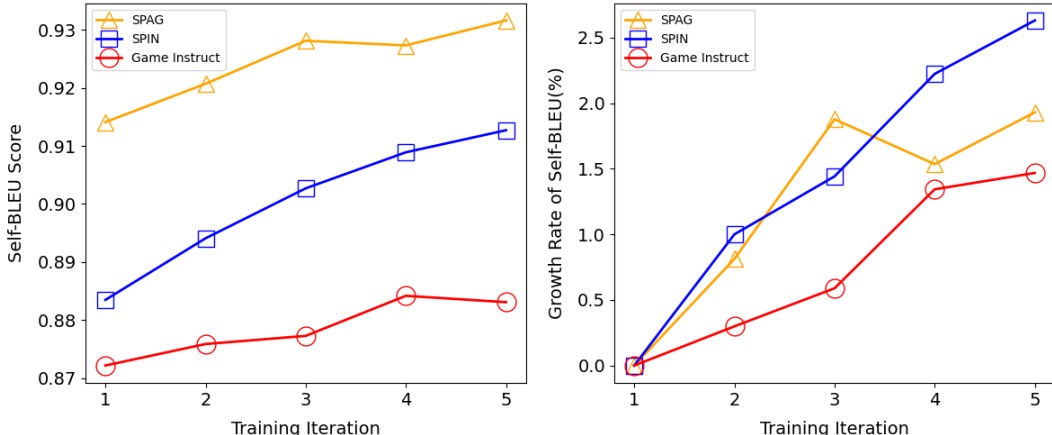

Figure 5: Diversity results of data in distinct self-play methods along with training iterations. The left figure represents the self-BLEU result in 2-gram setting while the right one exhibit the growth of average self-BLEU as the training proceeds

earlier model iterations and the golden, as well as comparisons between a model and its subsequent iterations. In addition, the debate type has garnered increasing attention recently. Cheng et al. (2024) implement the adversarial language game known as adversarial taboo (Yao et al., 2020), where an attacker and a defender engage in a conversation focused on a target word that is visible only to the attacker. Similarly, Ma et al. (2024) introduce the red-teaming game, in which large language models (LLMs) are initialized as a collective set of red-teaming policies designed to prompt the target LLM into generating harmful content. Expanding on this, Zheng et al. (2024) suggest forming an experiment where attakcer prompt a defender LLM with harmful content while the defender finds the weakness of it.

**Model Collapse** According to Alemohammad et al. (2023), in the context of a fully synthetic loop absent of sampling bias, both variational autoencoders (VAEs) and Gaussian mixture models yield mean absolute deviation (MAD) generative processes. Their findings indicate that both the synthetic augmentation loop and fresh data loop can lead to performance degradation in fine-tuned LLMs over successive generations. In a related study, Shumailov et al. (2024a) employs a guided diffusion model in the fully synthetic loop, and report that this approach mitigates declines in image quality. Furthermore, Martínez et al. (2023) demonstrate that a synthetic augmentation loop incorporating a Denoising Diffusion Implicit Model (DDIM) without sampling bias results in suboptimal performance over generations.

## 6 CONCLUSION

In this work, we propose a complex multi-player adversarial environment termed GAMEINSTURCT aimed at augmenting the reasoning abilities of LLMs by increasiing the complexity of self-play generated data during training. We apply the Chameleon Game for the LLMs to enhance their reasoning ability during the game. Besides, we also introduce dynamic reward to capture signals within conversations of players. Experimental results demonstrate that GAMEINSTRUCT can help improve the reasoning ability of LLMs while maintaining a stable training procedure against model collapse during self-iterative learning.

GAMEINSTURCT establishes a novel approach to augment the fundamental abilities of LLMs through self-play mechanisms. We hope future work can build on top of our multi-player environment by incorporating more sophisticated language game designs and a wider variety of task scenarios, to develop better models with the advanced capabilities.

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

## A    EVALUATION DETAILS

### A.1    REASONING BENCHMARK

We follow the setting of Open-LLM-Leaderboard as well as the Llama-2 paper, reporting 5-shot results for MMLU, 25-shot results for Arc, 0-shot results for TruthfulQA, 5-shot results for Winogrande, 5-shot results for GSM8k and 10-shot results for HellaSwag. The detailed descriptions of benchmarks are listed below:

- **MMLU** (Hendrycks et al., 2021) is a massive multi-task testset consisting of multiple-choice questions from various topics requiring the model to possess extensive reasoning and problem-solving ability as well as world knowledge

- **ARC** (Clark et al., 2018) is a comprehensive resource designed to evaluate the capabilities of models in scientific reasoning and problem-solving, featuring a diverse range of multiple-choice questions that span various scientific disciplines, thereby challenging models in their ability to understand and apply scientific knowledge through logical reasoning and common-sense inference.

- **TruthfulQA** (Lin et al., 2022) is a dataset specifically curated to assess the truthfulness of responses generated by language models, featuring a diverse collection of prompts that require accurate and reliable information, which test capabilities of models to understand the factual knowledge to reason the right answer

- **Winogrande** (Sakaguchi et al., 2019) is a large-scale dataset designed to evaluate the reasoning abilities of natural language processing models, comprising a wide array of ambiguous sentences that require contextual understanding and nuanced reasoning to disambiguate pronouns, thus serving as a robust benchmark for assessing a model's capacity to perform complex reasoning tasks in the realm of natural language understanding.

- **GSM8k** (Cobbe et al., 2021) is specifically created to evaluate the reasoning capabilities of language models in solving mathematical word problems, encompassing a diverse set of 8,000 examples that challenge models to understand contextual clues, perform arithmetic operations, and apply logical reasoning in order to derive correct answers, thereby providing a rigorous benchmark for assessing a model's proficiency in mathematical reasoning and problem-solving.

- **HellaSwag** (Zellers et al., 2019) is a dataset designed to rigorously test the reasoning abilities of language models by providing a collection of open-ended scenarios that require nuanced understanding and reasoning, featuring multiple-choice answers that demand models to predict the most plausible continuation of a given context, thus serving as a challenging benchmark for assessing a model's capability to perform complex reasoning and comprehend subtle contextual relationships in natural language.

## A.2 DIVERSITY CALCULATION

We adhere to establish practices recommended in Zhu et al. (2018), employing Self-BLEU score as the indicator of divergence. Self-BLEU initiate from BLEU, which aims to assess how similar two sentences are. Regarding one sentence as hypothesis and the others as reference, we can calculate BLEU score for every generated sentence, and define the average BLEU score to be the Self-BLEU of the document, the following is the process of calculating Self-BLEU:

$$\text{Self-BLEU} = \frac{1}{N}\sum_{i=1}^{N}\text{BLEU}(H_i, (H_1, H_2, ...H_i - 1, H_i + 1, ...H_n)), \tag{17}$$

where $N$ represents the total number of generated hypotheses $H_i$, and the latter one denotes the set of hypotheses excluding the $i$-th one.

We follow the approach suggested by Zhu et al. (2018) and calculate the average Self-BLEU score using n-grams ($n = 1, 2, 3, 4$), apart from that, to better represent the robustness of generated data during the self-iterative training process, we define growth of diversity as follows:

$$\text{Growth}_{\text{Self-BLEU}} = \sum_{i=1}^{N}\frac{\text{Self-BLEU}_i}{\text{Self-BLEU}_1} - 1, \tag{18}$$

where higher growth represents lower diversity during the self-iterative training process.

## B PROMPT FOR GAMEINSTURCT

> **Game Prompt**
>
> You are about to play the Chameleon Game. In this game, there are multiple players, one of them is chameleon while others are civilians. At the beginning, all the players are assigned a word, while the chameleon's word is different from the others.
>
> The game has two stages, conversation stage and voting stage. In conversation stage, each game is played in rounds. Each round consists of players describing the word they were given without directly revealing it. After everyone has spoken, the voting stage begins where players can vote each other based on the descriptions given. The aim is to vote who they think the chameleon is. The player with the majority votes is eliminated from the game.
>
> Here are the requirements:
> 1. All of the players are not allowed to directly say the word.
> 2. Players are not allowed to repeat or paraphrase descriptions by other players.
> 3. The description should only have information about specific words, don't generate any other sentence irrelevant to the word.
>
> The primary goal of the players bases on whether they are chameleon or civilians. The primary goal of chameleon is to avoid being identified. Thus it is expected to deduce the civilians' assigned roles and describe sentence. While civilians' primary goal is to notify other civilians of their true identity while prevent the potential chameleon to find out the real words

Figure 6: Chameleon Game rule prompt

**Player Prompt**

You are Player X. Your identity is Chameleon, your assigned word is {Word}. You are in the conversation stage. You are asked to describe the word, including its concept, usage, specialty and so on. You will be given the game history and your goal is to avoid being found out your identity by other players. Thus be aware of other players' description in the game history, combined with your assigned word to deduce their assigned word. Generating your description based on these deduction. Here are other requirements:
1.Do not pretend you are other players.
2.You don't need to prepend your player name to your response, despite others may do it.
3.You don't directly speak the word but describe it
4.Do not utter your thinking process
5.Your answer should only contain description around certain word, avoid generate answer containing any other information, such as your deduction about other players' assigned word

Game History: {history}
Response:

Figure 7: Chameleon Game Chameleon player prompt

**Player Prompt**

You are Player X. Your identity is Civilian, and your assigned word is {Word}. You are in the conversation stage. You are asked to describe the word, including its concept, usage, specialty and so on. You will be given the game history and your goal is to describe your word to notify other civilian of your identity but keep the potential chameleon unknown to your assigned word. Here are other requirements:
1.Do not pretend you are other players.
2.You don't need to prepend your player name to your response, despite others may do it.
3.You don't directly speak the word but describe it.
4.Do not utter your thinking process.
5.Your answer should only contain description around certain word, avoid generate answer containing any other information, such as your deduction about other players' assigned word.

Game History: {history}
Response:

Figure 8: Chameleon Game Civilian player prompt

