# OpenReview forum: "GameInstruct: Teaching Machines to Reason via Chameleon Game"
_ICLR.cc/2025/Conference — ICLR 2025 Conference Withdrawn Submission_

### Official Review · Reviewer_vMKr · 2024-10-19

**Soundness:** 3
**Presentation:** 3
**Contribution:** 3
**Rating:** 6
**Confidence:** 2

**Summary:**

This paper introduces a self-play method for generating synthetic alignment data called GAMEINSTRUCT.

This method employs the Chameleon Game to enhance LLM interactions and iteratively improve the capabilities of LLMs. A dynamic reward is designed for this scenario.

Extensive experiments are conducted to prove the effectiveness and potential of the proposed self-play method, including the potential of continuous improvement across training iterations, and robustness with respect to sampling temperature and model collapse.

**Strengths:**

- The proposed self-play method utilizing Chameleon Game shows its effectiveness by showing state-of-the-art performance on multiple benchmarks.
- The proposed method shows potential of continuous improvement across training iterations. Moreover, ablation experiments on self-BLEU score prove its robustness against model collapse compared to other self-play methods.
- The proposed Dynamic Reward Assigning method is proven to improve the performance of the authors' method on several benchmarks, and may generalize to other adversarial games.

**Weaknesses:**

- The idea that self-play adversarial games can be used for generating alignment data has been proven in some previous work, and the proposed method looks like replacing the old games with the Chameleon Game. While I recognize the contribution, strength and sate-of-the-art performance of this method, it would be more inspiring if the authors could provide more analysis or ablation experiments on why Chameleon Game is better than previously proposed games on generating synthetic data.
- The design of the dynamic reward looks generalizable to other adversarial games. However, effectiveness of it is mainly experimentally verified for Chameleon Game, but not for other adversarial games.

**Questions:**

The authors mentioned sophisticated language game designs with a wide variety of task scenarios for possible future work. Why Chameleon Game is better compared with previously proposed adversarial games like taboo in SPAG? What component of Chameleon Game makes it different?

---

> ### Author Response · Authors · 2024-12-03
> **Response to Reviewer vMKr**
>
> ## Question 1 & Weakness 1
> >  Why Chameleon Game is better compared with previously proposed adversarial games like taboo in SPAG? What component of Chameleon Game makes it different?
>
> > The idea that self-play adversarial games can be used for generating alignment data has been proven in some previous work, and the proposed method looks like replacing the old games with the Chameleon Game.
>
> Firstly, we note that the Language Taboo setup, where one party induces the other to say a specific word and the other party guesses the word from the induction, inherently reduces the diversity of data that can be generated by the defender. This task limits the defender's capacity to produce varied data. In contrast, the Chameleon Game, which involves both parties describing the word, excels in generating more diverse and complex data. By alternating different assigned words, we can inject knowledge from various domains into the model, enhancing its versatility.
>
> Furthermore, Language Taboo implicitly embeds the task of guessing the word within multiple rounds of dialogue, without explicit forms of thought such as chain-of-thought. This complexity is compounded as the defender must not only execute the generation task but also rely on their own responses for judgment. Conversely, the Chameleon Game simplifies the judgment process through subsequent voting, separating judgment from generation. This approach allows for a more focused generation process centered on describing the assigned word, providing the model with more precise instructions for corresponding generation, resulting in a greater diversity of data.
>
> Subsequently, regarding reward allocation, in the adversarial game of SPAG, rewards are assigned based on the round in which each utterance is made, rather than the specific content of the dialogue. This encourages the model to generate shorter dialogue turns and content, as this yields a greater reward. This is likely the reason for the observed performance decline in SPAG after just three rounds of iteration. In contrast, the Chameleon Game's reward distribution relies more on the specific content of each dialogue. As a result, responses that more easily reveal the chameleon's identity are discouraged, and the model is inclined to generate responses closer to those of the civilian's description. Our ablation study supports this, showing no significant increase in the self-bleu metric during iterative training, thereby demonstrating the effectiveness of our dynamic reward setting.
>
> Lastly, the Chameleon Game, in contrast to the two-person setup of the Taboo Game, employs a multi-player asymmetric game format. This requires defenders to generate distinct data, leading to a high degree of diversity in the first round of generated data. In comparison, SPAG exhibits the least diversity.

---

> > ### Comment · Reviewer_vMKr · 2024-12-03
> >
> > I would like to thank the authors for their rebuttals. I would like to keep my score.

---

### Official Review · Reviewer_5XZs · 2024-10-28

**Soundness:** 3
**Presentation:** 3
**Contribution:** 3
**Rating:** 5
**Confidence:** 3

**Summary:**

This paper proposes a self-play method called GAMEINSTRUCT, which leverages a multi-player game environment—specifically, the Chameleon Game—to improve language model reasoning by generating diverse, dynamic training data.  GAMEINSTRUCT incorporates multi-agent interactions with a dynamic reward mechanism. This mechanism assigns rewards based on individual player interactions rather than just game outcomes, enhancing the model's ability to develop reasoning skills.

GAMEINSTRUCT also utilizes imitation learning with data from advanced models like GPT-4 to enforce adherence to game rules, contributing to the model’s training effectiveness. Experimental results show that this approach significantly improves reasoning performance across benchmarks, maintaining stability and minimizing data redundancy over successive training iterations.

**Strengths:**

The strengths of GAMEINSTRUCT lie in its ability to enhance data diversity and model reasoning capabilities through a unique multi-player game-based self-play approach. It generates a broader range of interactions, reducing repetitive data and lowering the risk of model collapse. The incorporation of a dynamic reward mechanism, which evaluates player interactions rather than only final game outcomes, enables more refined training signals that boost the model’s reasoning skills. Additionally, experimental results demonstrate GAMEINSTRUCT’s effectiveness, with notable improvements in reasoning benchmarks and sustained stability across training iterations.

**Weaknesses:**

GAMEINSTRUCT might introduce higher computational demands due to multi-player interactions and a changing reward system, which may make it harder to scale for larger or limited-resource models. Additionally, it relies on imitation learning using data from advanced models like GPT-4, making it difficult to replicate without similar resources. The changing reward system, though helpful, adds complexity in setting accurate rewards, needing careful tuning for the best results. Finally, while effective for reasoning-based tests, it’s unclear if GAMEINSTRUCT performs well in other areas or tasks beyond language model reasoning.

**Questions:**

1, As mentioned in the weaknesses, how well does this approach scale when adding more agents? Can it handle the increase efficiently?

2, In section 3.3 on imitation learning, are you fine-tuning other LLMs using GPT-4 generated data? If so, why not use GPT-4 directly as an agent to play the game?

3, This method was only tested on the Chameleon game. Could you try applying it to other tasks as well?

---

### Official Review · Reviewer_4ZrJ · 2024-11-02

**Soundness:** 3
**Presentation:** 3
**Contribution:** 2
**Rating:** 3
**Confidence:** 4

**Summary:**

This paper introduces GAMEINSTRUCT, a novel training approach that enhances language models' reasoning capabilities through multi-player adversarial interactions using the "Chameleon Game" framework. The key innovation lies in addressing two major challenges in traditional self-play methods: insufficient data diversity and difficulties in reward signal design. In the Chameleon Game, multiple AI players interact where "civilians" share a common word while a "chameleon" must avoid detection while having a different word, creating complex dynamics that increase training data diversity and prevent model collapse. The authors also propose a dynamic reward algorithm that captures signals from player conversations throughout the game, moving beyond simple win/loss outcomes. Experimental results on the HuggingFace Open-LLM-Leaderboard demonstrate that GAMEINSTRUCT achieves notable improvements over existing self-play methods, particularly in reasoning tasks, while maintaining continuous improvement and data diversity during self-iterative training. The paper claims improvements of 1-2% across various reasoning benchmarks compared to state-of-the-art self-play methods, with the approach showing particular robustness against model collapse during extended training.

**Strengths:**

1. Leveraging the game playing to improve the reasoning capabilities is interesting.

2. The main contributions of this paper are i) the chameleon game, ii) the dynamic reward modeling, and iii) the RL training framework. Combining the three modules, the authors demonstrate that the reasoning capability of LLM can be improved.

**Weaknesses:**

1. The motivation of why solving games can improve reasoning capabilities is not very clear to me. There is no theoretical analysis about this.

2. This paper only considers a specific game. There are many games, that can also be potentially applied, by taking more games and more data into the training seems not much complexity will be introduced into the framework.

3. The improvement seems marginal.

**Questions:**

My questions are as follows:

1. Could the author provide theoretical justifications about why game playing can improve the reasoning capabilities of LLM? You employ the GPT-4 to generate the imitation learning, this may also improve the reasoning capability of LLMs? If yes, no game-playing is needed, just imitation learning. Even further, we can ask gpt-4 to solve complex decision-making tasks, and then generate the training data? therefore, still no game-playing is needed. How to justify this?

2. The improvement of this method seems marginal. How to justify that additional training with your methods is necessary, given that the improvement is small? Besides, compared with other SFT methods over high-quality training data, your method is much more complex. Therefore, how to justify the necessities of your method?

3. I also have one conceptual question. If game playing can really improve the reasoning capability of LLMs, does that mean the Nash Equilibrium strategy will be the most effective strategy to generate the training data? how about any other equilibrium concepts?

---

### Official Review · Reviewer_bmqi · 2024-11-04

**Soundness:** 3
**Presentation:** 2
**Contribution:** 2
**Rating:** 3
**Confidence:** 5

**Summary:**

The paper introduces GAMEINSTURCT, a novel approach within the domain of self-play for generating alignment data, which is crucial in reducing annotation costs during the alignment process. By leveraging a complex multi-player adversarial environment termed the "Chameleon Game," GAMEINSTURCT enhances the diversity of data generated during self-iterative training. This is achieved through multi-player interactions, which elevate the complexity and diversity of scenarios that a model encounters, thereby improving the model's reasoning abilities. Furthermore, the paper proposes a dynamic reward algorithm designed to capture nuanced signals within player conversations throughout the game, which aids in continuous performance optimization.

**Strengths:**

(1) The paper is articulate and well-organized, with clear definitions of key concepts and a logical flow of ideas. The use of the Chameleon Game as a case study helps in concretely demonstrating the application of GAMEINSTURCT, making the complex concepts more accessible to the reader.

(2) GAMEINSTURCT introduces a unique combination of a multi-player adversarial environment with a dynamic reward system tailored to self-play scenarios.

**Weaknesses:**

(1) It remains ambiguous whether GAMEINSTURCT is a game environment or a training method/framework. If GAMEINSTURCT includes a game environment, it would be beneficial for the community if the authors could open-source the code to allow for broader testing and adoption.

(2) The manuscript compares GAMEINSTURCT only with SPIN and SPAG, which seems like a narrow scope of comparison given the vast array of methods available that enhance data diversity and avoid local minima in self-play reinforcement learning before LLM emerges.
For example: population-based training, alphastar league training, PSRO, fictitious self-play, PFSP, CFR, and MCTS.

(3) The primary advantage of GAMEINSTURCT is highlighted as dynamic reward, which is widely known and used in reinforcement learning as a reward shaping technique. This raises concerns about the novelty of the proposed method.

(4) The experiments are only conducted in one environment, the Chameleon Game. There are numerous similar open-source environments, like werewolf, which could have been used to validate the findings more robustly.

(5) Sections such as 3.1 and 3.2 are overly verbose and could be condensed. The paper also contains obvious equations (e.g., eq13-eq16) that overshadow more critical details like RL training specifics, the number of imitation datasets used, how the dynamic reward evolved during training, and details on reward shaping. This lack of essential information diminishes the credibility of the work.

**Questions:**

(1) Can the authors clarify whether GAMEINSTURCT is intended as a game environment or a training method/framework? If it is an environment, are there plans to open-source the code?

(2) Broader Comparisons: Given the many existing techniques in self-play reinforcement learning, why were only SPIN and SPAG chosen for comparison? Could the authors consider broadening the scope of comparison to include more methods?

(3) Since dynamic reward is a well-understood concept in RL, can the authors discuss how their implementation of dynamic reward in GAMEINSTURCT provides a distinct advantage over existing methods?

(4) Are there plans to test GAMEINSTURCT in other environments beyond the Chameleon Game? This could help in understanding the robustness and generalizability of the proposed method.

(5) Could the authors provide more details on the RL training specifics, the size and source of the imitation datasets, the evolution of dynamic rewards during training, and the specifics of reward shaping? This information is crucial for evaluating the robustness and reproducibility of the results.

---

### Note · Authors · 2024-12-15

I have read and agree with the venue's withdrawal policy on behalf of myself and my co-authors.